# PIEZO1-Related Physiological and Pathological Processes in CNS: Focus on the Gliomas

**DOI:** 10.3390/cancers15030883

**Published:** 2023-01-31

**Authors:** Rui Hong, Dianxu Yang, Yao Jing, Shiwen Chen, Hengli Tian, Yang Yang

**Affiliations:** 1Department of Neurosurgery, Shanghai Sixth People’s Hospital Affiliated to Shanghai Jiao Tong University School of Medicine, Shanghai 200233, China; 2School of Medicine, Shanghai Jiao Tong University, Shanghai 200025, China

**Keywords:** PIEZO1, ion channel, mechanotransduction, glioma, glia cell

## Abstract

**Simple Summary:**

We summarized previous studies of PIEZO1 in neurons and glial cells of the central nervous system, and briefly reviewed its mechanotransduction mechanisms and its role in physiological and pathological processes. We further elaborated the effects of PIEZO1 in gliomas and its underlying mechanisms as well as its clinical application. Based on the existing advanced studies, we propose the promising potential of PIEZO1 in the treatment of neurological diseases, especially gliomas.

**Abstract:**

PIEZO1 is ubiquitously expressed in cells in different kinds of tissues throughout the body, which can sense physical or mechanical stimuli and translate them into intracellular electrochemical signals to regulate organism functions. In particular, PIEZO1 appears in complex interactive regulatory networks as a central node, governing normal and pathological functions in the body. However, the effect and mechanism of the activation or expression of PIEZO1 in diseases of the central nervous system (CNS) remain unclear. On one hand, in CNS diseases, pathophysiological processes in neurons and glial are often accompanied by variations in the mechanical properties of the cellular and extracellular matrix stiffness. The expression of PIEZO1 can therefore be upregulated, in responding to mechanical stimulation, to drive the biological process in cells, which in turns indirectly affects the cellular microenvironment, resulting in alterations of the cellular status. On the other hand, it may have contradictory effects with the change of active patterns and/or subcellular location. This review highlights the biological processes involved with PIEZO1 in CNS cells, with special emphasis on its multiple roles in glioma-associated phenotypes. In conclusion, PIEZO1 can be used as an indicator to assess the malignancy and prognosis of patients with gliomas, as well as a therapeutic target for clinical application following fully exploring the potential mechanism of PIEZO1 in CNS diseases.

## 1. Introduction

In physiological and pathological states, virtually all cells of our organism are stimulated at all times by a variety of mechanical forces arising intracellularly or persisting extracellularly, including matrix stiffness, tensile and fluid shear stress, and elastic deformations [1]. Sensing and appropriately responding to changes in the physical environment are essential for cells to perform their physiological functions. PIEZO1, a kind of ionic channel protein, was first identified and described in a mouse neuroblastoma cell line. It responds to mechanical stimuli and allows cations, such as calcium ions, to flow into cells [2]. PIEZO1 converts mechanical stimuli (strain) into biochemical signals, inducing alterations in protein conformation and activating intracellular biochemical signaling pathways. This process then specifically regulates the gene to drive cells to perform biological processes and molecular functions.

Maintaining a stable cytoskeletal structure and homeostasis of the extracellular matrix is the foundation for neurons and glial cells to perform physiological functions in the central nervous system (CNS) [3]. In the development of neurons and glial cells, PIEZO1 can not only drive development and differentiation, but also participate in the neuron-glial intercellular information exchange. These physiological functions among cells are essential for memory processes, cognitive functions, learning abilities, and motor capacities [4,5,6,7]. Several pathological processes, such as neuroinflammation, apoptosis, and tumor aggressiveness, are involved due to variations in the interacting forces between CNS cells and the extracellular matrix [8,9]. The role of PIEZO1 in other systemic cancers has inspired researchers to explore its use as a non-invasive ultrasound therapy in neurological diseases, such as gliomas [10,11,12]. Despite a proven role for PIEZO1 in the neuromodulation of the neuroblastoma cell line [13], its efficiency and safety in (pre-)clinical studies have not been measured to date.

## 2. Structure and Gating Mechanism of PIEZO1

With more than 2500 amino acids and a vast transmembrane domain, PIEZO1 is an evolutionarily conserved integral membrane protein that serves as a pore-forming subunit of the ion channel, but its structure and gating mechanism were not initially determined [14]. Due to the development of cryo-electron microscopy, live-cell immunoassays, and X-ray crystallography, its basic physical body has been identified and described as a homologous trimer assembled on a membrane with a helical blade-like structure [15,16]. It consists of a central cap, three peripheral blades, three long intracellular rays, an intracellular C-terminal domain (CTD), and a transmembrane (TM) fragment [15,16]. Based on the unique 38-TM structure inherent in its protomer, PIEZO1 can be divided into three functional components: 1. a mechano-sensing module for sensing mechanical stimuli, 2. a C-terminal ion-conducting pore module that acts as a channel, and 3. a transduction functional component for transmitting mechanical stimuli [17]. Each PIEZO1 protomer has an unprecedented structure of 38 TMs, of which the last 2 TMs at the C-terminus form an outer helix (OH) and an inner helix (IH), and the three IHs lock together to form a transmembrane pore [17,18]. The remaining TMs form abnormally curved, non-planar transmembrane helical blades that, all together, form nine tandemly repeating transmembrane helix units (THUs), where each unit consists of four TM fragments [18]. The partially non-planar TM blades form the central cap of the ion channel outside the cell, which is called the C-terminal extracellular structural domain (CED). The beam originates at the lower edge of the peripheral TM and physically connects the distal blade to the ion conduction pore module via a compound consisting of the CTD, anchor, and OH-IH. The specific structure of the beam reveals its dual role: transmembrane support and the transmission of conformational changes in extracellular blades [15,16]. The CTD, located above the proximal edge of the beam, interacts with the beam, and contains an anchor consisting of three alpha helices with a unique hairpin plane structure parallel to the film. The anchoring domain enables the formation of an inverted V-shape structure that penetrates the inner leaflet of the membrane, interacts with the IH and OH, and stabilizes the ion conduction pore [17].

It has been shown that PIEZO1 can be directly activated by the mechanical stretching of lipid bilayers [19,20]. The non-planar TM blades and THU structures form mechanoreceptors that directly sense minutiae changes in membrane curvature and tension [20,21]. The motion characteristics of the long rod-like beam and the peripheral THU are analogous to the lever motion, as the pivot on the beam near the central pore, consisting of residues L1342 and L1345, allows the PIEZO1 channel to efficiently translate the conformational changes of the extracellular blade into the opening and closing of the ion conduction pore, translating mechanical stimuli into chemical signals [18,22]. Based on the characteristic of PIEZO1 distorting the cytosolic membrane into a dome structure, some researchers have proposed the mechanism of “membrane doming” [23]. With the opening of the channel, the morphology of the membrane changes and the energy needed to modify the membrane is proportional to the change in the projected area under the dome [23]. This membrane doming mechanism explains the high mechanical sensitivity displayed by PIEZO1 through the ion selective pore.

Since membrane tension is largely dependent on the arrangement of the cytoskeleton and extracellular matrix (ECM), it is, therefore, reported that the cytoskeletal contractile protein myosin II stimulates PIEZO1 via the traction generated by myosin light-chain kinase (MLCK), and induces the change of the gate that triggers cation influx [24,25]. In addition, PIEZO1 also allows the interactions with a variety of proteins. These include myotubularin-related protein 2 (MTMR2) and stomatin-like protein3 (STOML3), which affect the activity of PIEZO1 channels according to cholesterol [26,27]. SERCA2 inhibits the PIEZO1 protein in a structural manner by binding to the anchored-OH junction of the linker pore module and the mechanical transduction module [28]. All in all, the PIEZO1 channel adopts both mechanotransduction and an ion channel to carry out its structural function in mechanosensing, channel opening, and ion conduction, which clarifies the channel mechanism based on mechanical force control.

## 3. Agonists and Antagonists

Many studies have investigated the ion conduction and mechanical gating mechanisms of the PIEZO1 channel. It has been reported that GsMTx4, an amphiphilic peptide toxin, may mechanically perturb PIEZO1 by modulating local membrane tension rather than directly acting on PIEZO1 itself [29]. Similarly, the amphiphilic macromolecule amyloid β (Aβ) can modulate channel activity by altering the membrane structure or cytoskeleton instead of working on it [30,31]. Ruthenium red (RR) works on the two residues with a negative charge in the intracellular CTD, blocking the ion conduction of PIEZO1 through a pore-plugging mechanism, while a series of TRPV channels other than the PIEZO1 channel are also inhibited by RR [14,22].

The cell membrane is composed of a lipid bilayer; saturated and polyunsaturated fatty acids can, therefore, regulate the activity of the PIEZO1 channel by adjusting membrane stiffness and lipids [32]. Compared with the antagonists of the PIEZO1 channels, Yoda1, the specific agonist of PIEZO1, is highly selective, acting exclusively on the variable binding domain of PIEZO1 [33]. As novel specifical activators of PIEZO1, Jedi1 and Jedi2 have been identified from more than 3000 compounds by using the fluorescent imaging plate technique as a calcium indicator [18]. Compared to Yoda1, Jedi1 and Jedi2 have higher water solubility, a faster onset current, and a shorter decay duration. More importantly, the joint application of Jedi1 and Yoda1 produces a synergistic effect of enhancing the inward current of PIEZO1 [18]. Recent studies have demonstrated that PIEZO1 is associated with various pathophysiological processes [34,35,36,37,38,39]. Drugs targeting it may, therefore, be promising for the treatment of many diseases [17,40,41,42]. However, the specific physiological mechanisms of its ligand-binding crystals still need to be further investigated since drugs that work on PIEZO1 lack robust specificity and potency [17]. Overall, with rapid progress in the elucidation of the structure and mechanism of PIEZO1, a growing number of agents are being developed for the treatment of PIEZO1 dysfunction-related diseases.

## 4. PIEZO1-Related Physiological and Pathological Processes

As an evolutionarily highly conserved protein, the PIEZO1 channel is expressed in a wide variety of cells that participate in different mechanotransduction processes under physiological and pathological conditions. The shear stress generated by the blood flow and membrane stretch caused by blood pressure fluctuations activates PIEZO1 channels embedded in the endothelium [43,44,45]. Several studies have shown that PIEZO1 regulates vascular remodeling, blood pressure, erythrocyte volume homeostasis, and lymphatic vessel development through Ca^2+^ influx or its interaction with integrin [44,45,46,47]. PIEZO1 has been shown to be overexpressed in osteoblasts as well as to affect the differentiation of osteoclasts and the resorption of bone by mediating the expression of type II and type IX collagen in osteoblasts that are governed by the YAP signaling pathway. In the case of PIEZO1 deletion, the osteoblast osteoclast crosstalk is out of balance, thus leading to the rapid loss of bone mass and causing spontaneous fractures [48,49].

Furthermore, the activation of PIEZO1 in myeloid cells initiates the effects of activator protein-1 (AP-1) and endothelin-1 (EDN1), thereby stabilizing the hypoxia-inducible factor 1α (HIF-1α) and inducing the expression of pro-inflammatory factors [50]. Another study suggests that toll-like receptor 4 (TLR4) works in synergy with PIEZO1 to strengthen the phagocytosis of macrophages to clear bacteria and resist the invasion of foreign substances [51]. Interestingly, PIEZO1 interacts with the classical inflammatory pathway JAK/STAT, the inflammasome NLPR3, the Ca^2+^-sensitive MAPK family, integrins, focal adhesion kinase (FAK), and calcium-dependent proteases to participate in the development and progression of inflammation [9,52,53,54,55,56,57,58,59,60]. PIEZO1 has been reported to be associated with iron metabolism in the organism, enhancing the ability of macrophages to engulf erythrocytes and inhibiting the expression of hepcidin, thereby leading to iron overload in the blood [61]. In radiation-injured endothelial cells of pulmonary microvascularization, PIEZO1 degrades cadherin in the vascular endothelium via calpain, causing an increase in ROS and the oxidation of lipids [62]. That is, PIEZO1 may also be involved in the regulation of endothelial cell ferroptosis.

In addition, PIEZO1 is overexpressed in a variety of tumors. The resting state of PIEZO1 has been reported to inhibit proliferation, migration, and invasion of tumor cells, and promote apoptosis of the cancer cells [9,11,12,34,63,64,65,66,67]. Mechanistically, the process is mediated mainly by an increase in the cytosolic calcium ion concentration, which initiates a series of relevant transductions of signals in the downstream, involving ERK1/2, AKT/mTOR, and YAP/TAZ [9,11,12,34,63,64,65,66,67]. To summarize, cells sense the mechanical stimuli via the PIEZO1 protein and then trigger calcium influx, which in turn triggers a cascade of downstream effects that ultimately induce conformational shifts of proteins and regulate gene expression, thus driving the cellular functions in physiological and pathological processes.

## 5. PIEZO1 in CNS Cells

Neurons and glial cells in the CNS exhibit significant mechanosensitivity, being capable of sensing mechanical stimuli from the ambient environment and converting them into biochemical signals to regulate their functions [19,36,68]. The critical value of PIEZO1 as a mechanosensor expressed in the membranes of the CNS is of growing interest, with an increasing number of studies aiming to discuss its role in neurological diseases [38,39,40,69]. As shown in Table 1, PIEZO1 is involved in the translation of mechanical signals in the CNS neurons and glial cells.

PIEZO1 is an essential ingredient in maintaining neuronal physiological function [4,5,6,75]. In a study measuring focal tissue stiffness in the brain at different developmental phases by observing brain tissue at different developmental nodes under an atomic force microscope, it was reported that PIEZO1 is fundamental in mentoring axonal growth and neurons’ maturation [6]. On the other hand, PIEZO1 channels can be activated to restrict nerve regeneration through the downstream cGMP kinase or PKG pathway in damaged neurons, triggering Ca^2+^ signaling [5,6,71]. Moreover, the mechanical activation of PIEZO1 with ultrasound is accompanied by an increase in Ca^2+^ influx and an elevated expression of activated calmodulin-dependent protein kinases (CaMKII) and the cAMP-response element binding protein (CREB) [70]. These two proteins are intimately implicated in neuronal plasticity, learning, and memory functions [70]. In addition, it is speculated that in the brain of those with Alzheimer’s disease (AD), astrocytes sense changes in the environment and subsequently relay the information to damaged neurons around the Aβ plaques instead of sensing stimuli via the damaged neurons that lost their PIEZO1 function [75]. Therefore, *PIEZO1* expression is not detectable in the neurons in damaged regions of the AD brain. In contrast, *PIEZO1* mRNA is detected in the neurons in both the non-AD brain and the non-damaged regions of the AD brain [75,76].

Astrocytes can take in and release a diversity of neuromodulatory signals for message delivery and account for the largest number of glial cells in the brain [77]. *PIEZO1* is expressed in reactive astrocytes surrounding plaques in AD patients, whereas *PIEZO1* mRNA is not detected in quiescent astrocytes, suggesting that PIEZO1 may be the expression gene in Aβ plaque-induced astrocytes [78]. Aging and infection can induce the upregulation of PIEZO1 in the plaque-induced reactive astrocytes, which may in turn trigger astrocyte proliferation [76]. In primary astrocytes derived from mice, some researchers use LPS to mimic the infection situation and observe an increase in PIEZO1 expression [79]. The upregulation of PIEZO1 elevates intracellular Ca^2+^ concentration while diminishing responsiveness of astrocytes to adenosine triphosphate (ATP), and inhibits the generation and release of pro-inflammatory cytokines and chemokines, ultimately suppressing neuroinflammation [79]. Moreover, evoked by external mechanical forces to generate cationic currents and Ca^2+^ signals, PIEZO1 in astrocytes also mediates spontaneous Ca^2+^ influx and triggers the release of ATP to regulate the development and neuronal maturation of neural stem cells (NSCs) [74]. In addition, the reduced volume of the hippocampal dentate gyrus in astrocyte-specific PIEZO1-deficient mice suggests that PIEZO1-mediated mechanotransduction affects the long-temporal enhancement (LTP) and the cognitive function of the hippocampus, consequently impairing learning and memory performance [74]. Consistent with this, the activity of PIEZO1 expressed on NSCs affects the definitive lineage selection of NSCs and guides their differentiation into either neurons or astrocytes [4]. Therefore, it is likely one of the mechanisms for the formation of astrocytes [4]. Collectively, the interaction between astrocytes and NSCs shows the significance of astrocytes as a center for intercellular message exchange in the CNS.

Under normal physiological conditions, the expression level of PIEZO1 in microglia is relatively high [80]. This may be because microglia, as the main residual immune cells in the brain, need to sense changes in the surrounding environment in time to perform immune functions for host defense [81]. Some studies have reported that any mechanical perturbation caused by intra- and extracellular osmotic pressure homeostasis or changes in environmental stiffness can activate PIEZO1 and provoke Ca^2+^ influx [72,73,80]. Elevated concentrations of intracellular Ca^2+^ interact with intracellular Ca^2+^-sensitive transcriptional regulators to regulate microglia proliferation and migration and to mediate neuroinflammation [72,80,82]. Immunofluorescence staining analysis of postmortem brain tissue sections from AD patients indicates that microglia cells cluster around Aβ plaques, and PIEZO1 protein levels are elevated [72]. In a recent study, it was proposed that microglia may sense the mechanical stimuli of plaques through PIEZO1 and enhance their own phagocytosis of foreign bodies, resulting in the cleavage, compaction, and later clearance of Aβ plaques [72]. Interestingly, the pathological effects of AD may modify the microglia membrane and cytoskeleton, impairing the PIEZO1-mediated Ca^2+^ signaling pathway and initiating downstream malfunction, which ultimately contributes to the inability of microglia to clear the Aβ amyloid plaques [80]. According to a new study using LPS to induce an inflammatory phenotype of microglia, the activation of the PIEZO1 channel is observed to inhibit the activation of microglia to exert anti-inflammatory effects, which suggests that microglia may regulate neuroinflammation via a novel mechanism involving PIEZO1 [8].

Recent studies have identified the expression of *PIEZO1* as a key effector of sensing the stiffness of the matrix in oligodendrocyte progenitor cells (OPCs) in rodents [7]. The differentiation and proliferation of OPCs are stopped in harder substrates, whereas in softer substrates or those inhibiting the PIEZO1, this phenomenon can be reversed, suggesting that OPCs are PIEZO1 dependent during growth [7]. Similarly, the expression level of PIEZO1 in human MO3.13 oligodendrocytes changes at different stages of maturation. When PIEZO1 is suppressed by the antagonist GsMTx4, it induces the enhanced proliferation and migration of oligodendrocytes [83]. Further investigation reveals that the condition of oligodendrocytes is related to neurons, as neuronal axonopathy is often accompanied by the demyelination of oligodendrocytes and vice versa [84]. This can be explained by the overexpression of *PIEZO1* channels in cortical neurons that leads to Ca^2+^ entering axons, triggering the release of more Ca^2+^ from the intracellular calcium reservoirs to activate the calpain-mediated demyelination [84].

Collectively, all these studies indicate that PIEZO1 mediates the glial-neuron interaction and plays a central role in the manifestations and pathological changes of brain function. Therefore, the specific mechanism needs to be clearly elucidated in future studies, as drugs that can target PIEZO1 for the neural regeneration and modulation of glial function have the potential to be used for the treatment of patients with brain diseases.

## 6. PIEZO1 in Gliomas

Gliomas, which originate from glial cells, are some of the most common brain tumors. The properties and behavior of the glial cells change during tumorigenesis [85,86]. Mechanical forces and biochemical signals control tumor formation and development during this process [85,86]. Strikingly, PIEZO1 physically localizes to the focal adhesion of glioma cells, catalyzing the maturation and growth of the focal adhesion through a force-dependent calcium signaling pathway [9,87]. A functional enrichment analysis of the China Glioma Genome Atlas (CGGA) dataset and the Gene Set Enrichment Analysis (GSEA) dataset revealed that PIEZO1 acts as a central node in a functional regulatory network that integrates regulators associated with tissue-stiffening molecules [63]. Consistent with this, RNA sequencing of PIEZO1 knockout glioblastoma cell lines and the TCGA database analysis for double-determining PIEZO1-related genes show that PIEZO1 is associated with the extracellular matrix (ECM), actin cytoskeleton remodeling, and the activation of integrin adhesion signaling [9]. In addition, the expression of PIEZO1 transduces mechanical stimuli among the glioma cells, promoting tumorigenesis and development [9]. This finding can explain how the changes in self-stiffness in gliomas are mainly caused by the pressure gradient generated by the tumor itself instead of collagen deposition or cross-linking [88]. The enhanced stiffness among glioma cells and the ECM microenvironment in turn regulates the activation of PIEZO1, facilitating the pathological process [9,87,88]. Moreover, many signaling pathways, including the matrix metalloproteinase (MMP) family, tissue inhibitors of the metalloproteinases (TIMP) family, the mitogen-activated protein kinase (MAPK) family, and the phosphoinositide 3-kinase (PI3K) family, are positively associated with the higher expression of PIEZO1 during the pathological process [63]. In short, an increased expression of PIEZO1 in most aggressive tumors, including glioblastoma, indicates a mechanical sensing and growth advantage of glioma cells.

Clinical studies on the expression of PIEZO1 in patients with gliomas demonstrate a similar tendency. The analysis of a clinical database that consists of 325 gliomas cases from the CGGA dataset and 276 cases from the GSE16011 cohort showed that the expression of PIEZO1 is highly correlated with the malignancy and molecular subtypes of gliomas [63]. Moreover, the overexpression of PIEZO1 contributes to more severe clinical symptoms, as was found via a retrospective analysis of imaging data and surgical samples from 64 patients with glioblastoma [89]. Additionally, immunohistochemical analysis of PIEZO1 in 183 patients with gliomas suggested that PIEZO1, as an independent factor, has an adverse impact on the prognosis of glioma patients [90]. The combination of the PIEZO1 expression level and WHO grade is much more accurate in predicting clinical outcomes [90]. Overall, PIEZO1 can be used as an indicator of glioma malignancy and is able to predict the clinical outcome in patients with gliomas.

In addition, peritumoral brain edema (PTBE), including vasogenic and cytotoxic edema, exacerbates neurological signs and clinical symptoms in patients with gliomas, which can serve as an independent factor for predicting the prognosis and recurrence of the gliomas in such patients [89,91,92]. It has been postulated that vasogenic edema can be mediated through Ca^2+^ influx by opening the PIEZO1 ion channels, and then activating calpain and degrading the tight-junction protein between adjacent cells in glioblastomas (Figure 1a). There is, however, a lack of evidence for the further validation of this hypothesis [89]. Theoretically, cytotoxic edema happens due to extracellular cations entering neurons and glial cells through cation channels and accumulating in cells while cationic influx drives anions inflow [92]. The PIEZO1 ion channels may play a vital role during this process. However, no studies have yet supported the relationship between PIEZO1 and cytotoxic edema.

However, the emerging evidence shows that the activation of PIEZO1 may be a sonodynamic therapeutic target [10]. Since it helps transient Ca^2+^ influx combine with the lipid droplets, it forms a complex that disturbs the energy supply in gliomas in addition to cell swelling lead by the calcium pathway [10]. Together with previous studies, the opposite effect, led by the activation of PIEZO1, may be related to different mechanisms. On one hand, during glioma genesis, PIEZO1 gathers around focal adhesions, which activates regional calcium fluctuations and leads to adhesion maturation and cell polarization [87]. In addition, PIEZO1 interacts with integrin-dependent kinases (FAKs) and transmits signals to the transcriptional coactivator with a PDZ-binding motif (TAZ), leading to chromatin remodeling and changes in transcription levels (Figure 1b). On the other hand, as a therapeutic target in gliomas, PIEZO1 acts as an ion channel for translating mechanical stimuli to electrical and chemical signals. The mild calcium influx would change to a large intracellular calcium transient stimulated by ultrasound, leading to a drastic change in the voltage in the tumor cells (Figure 2).

## 7. Conclusions and Perspective

Neurons and glia cells are exposed to and respond to different types of stimuli, displaying high sensitivity in the CNS [19,68]. The study of mechanotransduction pathways is an emerging field in the treatment of neurological diseases. The recently identified mechanosensitive ion channel PIEZO1 shows distinct properties in the development of neurological disorders, particularly in its interactions with intracellular soluble signals, which may help identify potential therapeutic targets [36]. In this review, we summarized the role of PIEZO1 in neurons and glial cells and described the pioneering research in gliomas.

PIEZO1-mediated cellular responses have been reported to exhibit contradictory phenotypes, such as anti- versus pro-inflammatory and pro-regenerative versus regenerative inhibition, but the most important thing is the confirmation of the ability of PIEZO1 to move the phenotype of glial cells in the required direction [8,51]. Similarly, PIEZO1 has been found to promote tumor aggression and development [9]. It is also reported to act as a mediator of tumor apoptosis in gliomas [10]. This can be theoretically explained as it may depend on the subcellular localization of PIEZO1 and the way PIEZO1 is activated [9,87]. Traction on the local plasma membrane and cytoplasm induces cell division, whereas extrusion on the local cytoplasm forms cytoplasmic aggregates and induces apoptosis [9,65]. In the absence of the contact inhibition of tumor cells in glioma development, the mechanosensory function of PIEZO1 is reconnected with the unidirectional ability to promote malignant tumor progression [9].

Although the clear relationship and mechanisms between PIEZO1 and neurological diseases such as AD, Parkinson’s disease, stroke, and spinal cord injury are still unclear, a rising number of studies are gradually uncovering the underlying mechanisms involved [3,36,40,41,93]. With further research on PIEZO1, we can leverage the potential anti-inflammatory effects of astrocytes and microglia to restore mechanotransduction pathways and neuron-glia crosstalk homeostasis around amyloid plaques by using PIEZO1 as a new drug target for neurodegenerative diseases and CNS injury [8,76,79]. One of the challenges of neural stem cell transplantation therapy is determining how to best direct cell differentiation after transplantation, and PIEZO1 plays a pivotal role in the spectral specification of neural stem cells [4]. Drugs that target and regulate the activity of PIEZO1 may therefore provide opportunities for neural stem cell transplantation therapy.

In addition to neurodegenerative diseases, gliomas are some of the most common brain tumors [85,86]. Several studies have explored the use of PIEZO1 as an indicator of tumor malignance and prognosis [63,89,90]. A recent study has reported a linear correlation between the expression level of PIEZO1 and the severity of peritumoral edema [89]. Edema is postulated to be caused by the opening of the blood–brain barrier (BBB), leading to an influx of calcium and the activation of calpain that degrades the tight junctions of the BBB [89,94]. The inhibition of PIEZO1 may, therefore, reduce clinical symptoms in patients with gliomas and can be a therapeutic target for diseases involving the collapse of the BBB [89]. As a non-selective cation channel, PIEZO1 is located in both plasma and endoplasmic reticulum membranes. Once opened, it will lead to the accumulation of cytotoxic Ca^2+^ in the cytoplasm and mitochondria, which, in combination with chemotherapy and radiotherapy, promotes the apoptosis of cancer cells [95,96]. Previous studies have suggested that it is feasible to treat glioblastoma with autologous T-cell therapies expressing chimeric antigen receptors (CARs). However, the off-target effect, caused by the tumor microenvironment, tumor heterogeneity, and the antigenic loss of the glioblastoma, makes it difficult to accurately kill the glioblastoma [97]. We can attempt to achieve controllable, non-invasive, and precise immunotherapy for gliomas by expressing PIEZO1 on T-cells, and, subsequently, by using ultrasound or drugs targeting PIEZO1 to locally stimulate gliomas, thereby activating PIEZO1 and inducing the activation of the nuclear factor of the T-cell (NFAT) that drives the expression performance of target genes [98].

In summary, PIEZO1, as a rising star molecule, can guide the functional phenotypes of neurons and glial cells, and participate in a variety of neurological diseases. In particular, PIEZO1 can be used as an indicator to assess the malignancy and prognosis of patients with gliomas, as well as a therapeutic target to control tumor progression. Specific mechanistic studies centered on PIEZO1 will contribute to our understanding of the mechanobiology of CNS diseases and help us develop new therapeutic approaches for patients with gliomas.

## Figures and Tables

**Figure 1 cancers-15-00883-f001:**
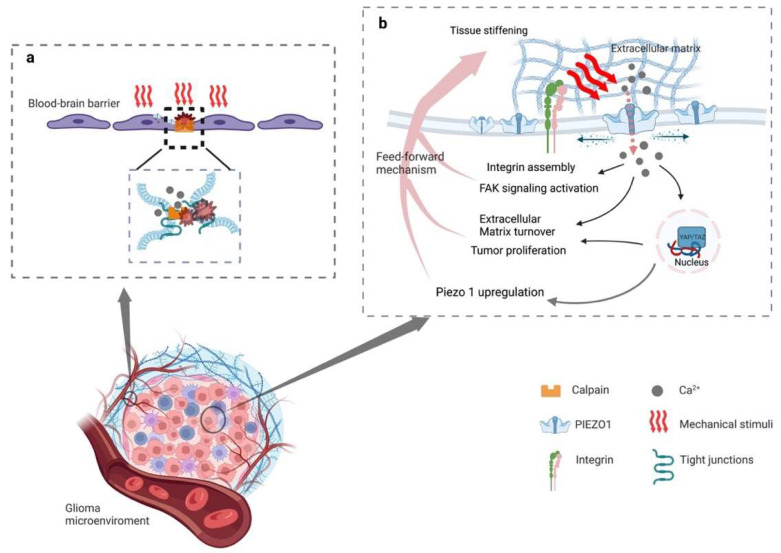
Schematic diagram of the mechanism of activated PIEZO1 in glioma progression and peritumoral edema. (**a**) Stiffening of the extracellular matrix activates PIEZO1, resulting in an influx of calcium. Intracellular elevated concentrations of Ca^2+^ catalyzes the assembling and maturation of focal adhesion, and (in-)directly activates integrin-focal adhesion pathways that stimulates cell proliferation and regulates extracellular matrix remodeling. The increased expression of PIEZO1 reinforces the mechanosensory and mechanotransduction capacity of the tumor cells, generating a feedforward mechanism that promotes glioma progression. (**b**) Proliferating tumor cells compressing peripheral blood vessels and activating the PIEZO1 of endothelial causing calcium influx. The influx of calcium then reacts with calpain, which disrupts the tight junctions between the blood-brain barrier.

**Figure 2 cancers-15-00883-f002:**
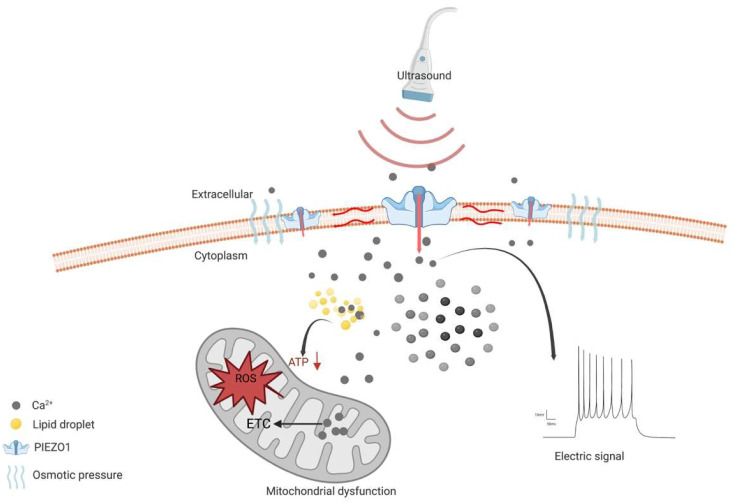
Schematic diagram of the mechanism of PIEZO1 as an anti-glioma therapeutic target. PIEZO1 is activated by ultrasound and constantly opens, leading to Ca^2+^ influx and increased concentration of intracellular Ca^2+^. The osmotic pressure between intracellular and extracellular leads to cell swelling and induces cell death. High concentration of cytoplasmic Ca^2+^ enters mitochondria and elevates intra-mitochondria Ca^2+^, resulting in impaired electron transport chain (ETC) function. Subsequently, impaired ETC leads to the elevation of reactive oxygen species (ROS). All above are contributed to mitochondrial dysfunction, leading to cell death. In addition, the instantaneous opening of PIEZO1 leads to the rapid transmembrane of cations and a rapid increase in intra- and extracellular voltage difference.

**Table 1 cancers-15-00883-t001:** Mechanotransduction of PIEZO1 in CNS cells.

Cell Type	Mechanical Stimuli	State of PIEZO1	Effects	References
Neuron	Ultrasound	activated	Elevated intracellular Ca^2+^	[13]
		activated	Initiating Ca^2+^ influx and affecting the levels of downstream Ca^2+^ signaling proteins involved in neuronal function	[70]
	Substrate stiffness gradient	activated	Axonal growth and pathfinding errors	[6]
	Axon injury	activated	Inhibiting axon regeneration via the CamKII-Nos-PKG pathway	[5]
	Oxygen-glucose deprivation/reoxygenation injury	activated	Enhanced cell viability inhibition, apoptosis, increase intracellular calcium levels and enhanced calpain activity	[71]
Microglial	Amyloid beta fibrils stiffness	activated	Inducing Ca^2+^ influx, phagocytosis and compacting of Aβ plaques	[72]
	Osmotic pressure	activated	Increasing cytosolic Ca^2+^ signaling and regulate cell function via JNK1 and mTOR signaling pathway	[73]
Astrocytes	Mechanical indentation stimulation	activated	Evoking Ca^2+^ response and ATP release as therefore regulates neurogenesis and cognitive functions	[74]
Oligodendrocyte progenitor cells	Mechanical stiffness gradient	inhibited	Increasing proliferation and differentiation	[7]
Neural stem cells	Stretch stress	activated	Directing the fate of the neural stem cells toward the desired lineage.	[4]

Abbreviations: CamKII: calmodulin-dependent protein kinases; Nos: nitric oxide synthase; PKG: cGMP-dependent protein kinase G; JNK1: c-Jun N-terminal kinase 1; mTOR: mammalian target of rapamycin; ATP: adenosine triphosphate.

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
