# Peer review of "PIEZO1-Related Physiological and Pathological Processes in CNS: Focus on the Gliomas"

_cancers, 2023, doi:10.3390/cancers15030883_

Round 1
Reviewer 1 Report
This manuscript summarizes recent trend in study of PIEZO1 in terms of pathology, which is very informative and elucidating.
There are a few improvements to be applied.
[Clarification of sentanses/paragraphs]
1. 226. Therefore, it is likely one of the mechanisms for the proliferation of astrocytes.
~according to the cited reference it's written,
'we find that stiff substrates increase Piezo1 activity and favor neuronal specification of hNSPCs, whereas inhibition of Piezo1, either by genetic knockdown or by pharmacological inhibition, has the opposite effect and reduces neuronal specification'
Line 226 doesn't seem to reflect the true meaning of the cite reference like above so need to be remodified.
2. 282~296 paragraph describes the mechanism of tumorigenesis of PIEZO1. However, it sounds superficial or circular so needs to address in more detail and succinct way.
[Grammatical issues]
1. 101 As the membrane tension depends largely on the arrangement of the cytoskeleton and extracellular matrix (ECM).
2. 238 microglia are clustered around Aβ plaques and elevated levels of PIEZO1 protein
3. 264 plays a central role in the performance brain functions and pathological changes.
4. 266 regulating functions of glial with drugs
5. 288 The increasing stiffen among the abundant glioma cells
6. 295 mechanicalsensing
7. 358 Neurons and glia cells are exposed to different forces and respond to different kinds of stimuli accordingly with high sensitivity in the CNS
8. 384 Among the difficulties of neural stem cell transplantation therapy is determining how to best guide cell differentiation after transplantation with PIEZO1 playing a pivotal role in the spectrum specification of neural stem cells
9. 405 makes it difficult to achieve precise killing in glio-blastoma
Author Response
We want to extend our appreciation for taking your time and effort to provide such great guidance. We carefully considered your comments:
- 226. Therefore, it is likely one of the mechanisms for the proliferation of astrocytes.
~according to the cited reference it's written: 'We find that stiff substrates increase Piezo1 activity and favor neuronal specification of hNSPCs, whereas inhibition of Piezo1, either by genetic knockdown or by pharmacological inhibition, has the opposite effect and reduces neuronal specification'
Line 226 doesn't seem to reflect the true meaning of the cite reference like above so need to be remodified.
Answer: Our conclusion “it is likely one of the mechanisms for the proliferation of astrocytes” comes from the following sentences in the cited reference:
- “Inhibition of channel activity by the pharmacological inhibitor GsMTx-4 or by siRNA-mediated Piezo1 knockdown suppressed neurogenesis and enhanced astrogenesis.”
- “Thus, pharmacological inhibition of SACs appears to promote astrocyte formation and to reduce neuron formation.”
- “These results recapitulate the effect of pharmacological SAC inhibition on lineage choice and demonstrate that Piezo1 activity is involved in neuronal–glial specification of hNSPCs.”
We therefore conclude that “the activation of PIEZO1 may be a mechanism for the proliferation of astrocytes”. Now we reconsider the wording and replace “proliferation” with “formation”.
- 2. 282~296 paragraph describes the mechanism of tumorigenesis of PIEZO1. However, it sounds superficial or circular so needs to address in more detail and succinct way.
Answer: Thank you for your valuable comment. We have rewritten this paragraph and merge it with the previous paragraph as “In addition, the expression of PIEZO1 transduces mechanical stimuli among the glioma cells promoting tumorigenesis and development [9]. This finding can explain that ……”.
- [Grammatical issues]
- 101 As the membrane tension depends largely on the arrangement of the cytoskeleton and extracellular matrix (ECM).
- 238 microglia are clustered around Aβ plaques and elevated levels of PIEZO1 protein
- 264 plays a central role in the performance brain functions and pathological changes.
- 266 regulating functions of glial with drugs
- 288 The increasing stiffens among the abundant glioma cells
- 295 mechanicalsensing
- 358 Neurons and glia cells are exposed to different forces and respond to different kinds of stimuli accordingly with high sensitivity in the CNS
- 384 Among the difficulties of neural stem cell transplantation therapy is determining how to best guide cell differentiation after transplantation with PIEZO1 playing a pivotal role in the spectrum specification of neural stem cells
- 405 makes it difficult to achieve precise killing in glioblastoma
Answer: It is true that grammatical issues are a significant obstacle to the readability of our academic manuscript. Therefore, we have rewritten many sentences, including those mentioned above, to make them clearer, and use the MDPI's “English Editing Service” to improve the readability of our manuscript.
We hope these explanations and changes to our manuscript meet your expectations to publish and look forward to hearing from you. Please find attached the revised paper with track changes indicating all revisions.
Sincerely,
Corresponding author:
Yang Yang, MD, PhD
Department of Neurosurgery,
Shanghai Sixth People's Hospital Affiliated to Shanghai Jiao Tong University School of Medicine
E-Mail: yang.cne.yang@gmail.com

Reviewer 2 Report
The manuscript is well-written, and the contents are well-organized and explained scientifically. This manuscript will be very helpful to those trying to research PIEZO1. The below issues should be carefully addressed. The content of the table size and the table border line should be checked carefully. And, English needs careful revision, there are many grammatical and typing mistakes(superscript & word). I would recommend it for publication in the Cancers.
Author Response
Thank you so much for your valuable comments. We carefully considered the comments and changed parts of our manuscript accordingly:
This manuscript will be very helpful to those trying to research PIEZO1. The below issues should be carefully addressed. The content of the table size and the table border line should be checked carefully. And, English needs careful revision, there are many grammatical and typing mistakes (superscript & word). I would recommend it for publication in the Cancers.
Answer: Thank you for your appreciation and interest in our manuscript. The table does not look beautiful due to the layout. We have changed the layout, now it looks much better. Regarding the grammatical and typing mistakes, we have rewritten many sentences, including those mentioned above, to make them clearer, and use the MDPI's “English Editing Service” to improve the readability of our manuscript.
Thank you again for your time and effort.
We hope these explanations and changes to our manuscript meet your expectations to publish and look forward to hearing from you. Please find attached the revised paper with track changes indicating all revisions.
Sincerely,
Corresponding author:
Yang Yang, MD, PhD
Department of Neurosurgery,
Shanghai Sixth People's Hospital Affiliated to Shanghai Jiao Tong University School of Medicine
E-Mail: yang.cne.yang@gmail.com
